

# Fatbox: the fault analysis toolbox

Pauline Gayrin[1,2], Thilo Wrona[3], Sascha Brune[1,2], Derek Neuharth[1], Nicolas Molnar[4], Alessandro La Rosa[5], John Naliboff[6]

[1] GFZ Helmholtz Centre for Geosciences, Potsdam, 14473, Germany
[2] University of Potsdam, Potsdam, 14476, Germany
[3] Deutsche ErdWärme GmbH & Co KG, Karlsruhe, 76133, Germany
[4] Formerly at University of Aachen, Aachen, 52072, Germany
[5] University of Pisa, Pisa, 56126, Italy
[6] New Mexico Institute of Mining and Technology, Socorro, NM 8780, United States of America

*Correspondence to*: Pauline Gayrin (gayrin@gfz.de)

**Keywords:** automated mapping, feature extraction, normal faults, structural analysis, fault networks, digital elevation models, numerical modelling, analogue modelling.

**Abstract.** Understanding complex fault networks is essential for reconstructing their geological history, quantifying deformation in tectonically active regions, and assessing geohazards and resource potentials. Structure and evolution of fault networks are investigated using a range of methods, including numerical and analogue modelling, as well as the analysis of topographic data derived from satellite imagery. However, due to the high density and complexity of fault systems in many study areas or models, automated analysis remains a significant challenge, and fault interpretation is often performed
manually. To address this limitation, we present Fatbox, the fault analysis toolbox, an open-source Python library that integrates semi-automated fault extraction with automated geometric and kinematic analysis of fault networks. The toolbox capabilities are demonstrated through three case studies on normal fault systems: (1) fault extraction and geometric characterization from GLO-30 topographic data in the Magadi-Natron Basin; (2) spatio-temporal tracking of fault development in vertical cross-sections of a forward numerical rift model; and (3) surface fault mapping and geometric
evolution of an analogue rift model. By representing fault networks as graphs, Fatbox captures the complexity and variability inherent to fault systems. In time-dependent models, the toolbox enables temporal tracking of faults, providing detailed insights into their geometric evolution and facilitating high-resolution measurements of fault kinematics. Fatbox offers a versatile and scalable framework that enhances the efficiency, reproducibility, and precision of fault system analysis—opening new avenues for tectonic research.



## 1 Introduction

Faults and fractures are discontinuities within rock volumes that form when applied stresses exceed the mechanical strength of the host rock, leading to brittle failure. These structures occur across a vast range of spatial (from micrometres to kilometres) and temporal (from years to millennia) scales (Scholz, 2019), often evolving into intricate, highly complex networks (Osagiede et al., 2023; Tewksbury et al., 2014). Fault networks accommodate deformation in many other geological environments, and their architecture evolves over time as individual faults grow, interact, and sometimes link together. This growth and interaction process plays a crucial role in shaping the fault network's geometry and mechanical behaviour. Geometric analyses of fault structures can therefore be used to estimate regional strain and, where sufficient temporal constraints exist, to quantify strain rates - analyses that are mostly conducted in continental rifts where normal fault networks are identifiably from satellite data (La Rosa et al., 2025; Polun et al., 2018; Riedl et al., 2022). At plate boundaries, faults are fundamental in accommodating strain, and detailed characterization is essential for evaluating the contribution of distributed, off-fault deformation to the total slip budget (Herbert et al., 2014). Beyond their mechanical role, faults significantly influence the rheological and transport properties of the crust by serving as conduits for fluids, volatiles, and deeply sourced gases (Frondini et al., 2008; Muirhead et al., 2020; Tamburello et al., 2018). In continental rifts, fault activity frequently alternates with magmatic processes, controlling the localization of permeability and geothermal activity (Corti, 2012; Jolie et al., 2021). Mechanical studies from rift basins further suggest that large normal faults commonly originate through the progressive growth and linkage of smaller-scale faults (Faulkner et al., 2010; Rotevatn et al., 2019), highlighting the importance of understanding fault network evolution when assessing the temporal evolution of rift basins.

Fault networks can be investigated using a range of complementary methods, each offering a different perspective on deformation processes: (1) Mapping of natural fault systems combines various remote sensing techniques and geophysical methods to identify and characterize faults at the surface and at depth. Geological mapping focuses on the surface expression of faults (Muirhead et al., 2016), while seismic data helps detect subsurface displacements along stratigraphic horizons (Wrona et al., 2023). Magnetotelluric surveys further contribute by mapping variations in electrical resistivity, which can indicate the presence of fluids or fault zones at depth (Martí et al., 2020), and can be used in conjunction with satellite imagery. Normal faults, in particular, are well-suited for remote sensing approaches due to their strong topographic signature in extensional settings. These faults can be detected either by identifying topographic shifts caused by displacement—using resources such as the Copernicus GLO-30 dataset, an open-source global topographic model released by the European Space Agency in 2019—or by recognizing their traces in optical imagery (La Rosa et al., 2022). Unlike thrust faults, which often collapse the hanging wall in compressional settings, normal faults produce first-order topographic features that make them especially visible in satellite data. (2) Numerical modelling provides a powerful tool for simulating geodynamic processes over spatial and temporal scales relevant to Earth's evolution. These models incorporate well-constrained rheological properties of Earth's layers (Bürgmann and Dresen, 2008), simulating deformation through brittle failure using plasticity



(Davis and Selvadurai, 2002) or ductile flow based on experimental rheology laws (Hirth and Kohlstedt, 2004). Depending on the research focus, models may address fault evolution in crustal-scale setups (Allken et al., 2011) or include the mantle
to examine long-term rift evolution and the onset of seafloor spreading in extensional systems (Li et al., 2024). Although often conducted in two dimensions, such simulations offer high spatial resolution, enabling detailed exploration of fault geometries and their temporal development (Neuharth et al., 2022). (3) Analog models simulate natural processes under controlled conditions, using carefully chosen setups and materials to accurately reproduce key aspects of the target system. To elucidate normal fault systems, analogue models typically focus on the shallow crust, aiming to replicate fault network
geometries and deformation patterns (Maestrelli et al., 2024; Zwaan et al., 2021).

At present, fault and fracture systems are primarily mapped manually using field, satellite, and geophysical data (Claringbould et al., 2020; Muirhead et al., 2016). Manual faults interpretations performed by expert geologists and corroborated by field observations are the most common and reliable methods (La Rosa et al., 2022; Muirhead et al., 2016).
However, this approach demands substantial expertise and time, and mapping consistency is not always ensured, especially in large areas (Bond, 2015). Semi-automated workflows can perform high-resolution fault mapping using regional DEMs (Ahmadi and Pekkan, 2021), though these methods necessitate careful calibration against manual interpretations to ensure accuracy. Recently, several semi-automated fault detection methods have been developed (Healy et al., 2017; Mattéo et al., 2021; Wrona et al., 2021; Zielke et al., 2010). Certain approaches focus on the quantification of lateral and vertical
displacements by analysing geomorphological offsets, such as river channel shifts (Stewart et al., 2018; Zielke et al., 2012). Other methods employ U-Net convolutional neural networks for the automatic detection of fractures and faults in optical images and DEMs Mattéo et al. (2021). Existing tools are typically designed for a specific task, such as network extraction from images (Dirnberger et al., 2015), topological analysis (NetworkGT by Nyberg et al., 2018), remote sensing data processing (Ahmadi and Pekkan, 2021), fracture system analysis (FracPaq by Healy et al., 2017), or automated throw
calculation (Auto_Throw by T et al., 2025). However, so far, no tool unites these functionalities within an open-source framework that allows multiple data source types, temporal correlation, and both geometric and kinematic fault descriptions.

Here we introduce Fatbox - the Fault Analysis Toolbox, an open-source Python software that includes a comprehensive suite of functions for extracting and analysing faults and fractures in various observational datasets and models. In Fatbox, faults
are described as networks that capture the spatial evolution of the systems, which allows for the representation of rich and complex geometries as observed in natural fault networks. Additionally, the toolbox includes functions for correlating faults over time, allowing for the tracking of the system's temporal evolution. Furthermore, classical geometric fault analysis parameters like length, throw, and displacement (Fossen, 2016) have been incorporated to quantify and visualize fault network evolution. In this study, we present the different functionalities of Fatbox using three application examples that best
illustrate its potential: (1) Fault extraction and geometric analysis of a topographic dataset at the example of the Magadi Natron Basin. 2) Fault extraction and temporal tracking using depth profiles from a numerical rift model. (3) Fault mapping





and fault geometry tracking within an analogue model. Each application serves to illustrate specific functionalities of the toolbox, addressing distinct analytical requirements. The study concludes with a discussion of possible future developments.

## 2 Background and technical setup

### 2.1. Definitions and graph description

Fatbox extracts and quantifies the geometry of fault networks, accounting for their complex spatial and temporal evolution. Developed in Python 3.9, Fatbox provides functions for mapping, editing, analysing, and tracking the temporal evolution of fault networks. Two types of input data are typically used: topography or strain. The fault extraction can be conducted semi-automatically or imported in the workflow from an existing dataset of manually picked faults.

In Fatbox, the fault systems are described as networks. Each fault extracted is called a component, and is assigned to an identification number referred to as the label. Each component of the network consists of nodes (points defined by location: x- and y-coordinates) and edges (connections between nodes) (Fig 1 B-D). This network-based representation allows us to capture connections observed in natural fault systems, such as fault splays (nodes with three edges) and intersections (nodes with four edges) (Fig 1 C-D). The key functionalities of the toolbox are 1) automated fault extraction from various datasets, including top-view and depth sections; 2) temporal tracking of structures; 3) measurement and quantification of multiple structural and geometric parameters across the fault network. The fault networks are described as graphs (NetworkX, (Hagberg et al., 2008)) and can be exported in raster, shapefile, and KMZ formats. The geometric measurements are systematically stored within the graphs and are readily accessible throughout the workflows.

Fatbox comprises six Python module files and three Jupyter notebook tutorials. The modules organize the functions by specific tasks and are callable within separate scripts. The tutorials illustrate the main applications of the toolbox as detailed in this study: elevation data of a natural rift, numerical modelling, and analogue modelling. Each tutorial contains multiple Jupyter notebooks that outline key functionality step-by-step and explains the corresponding workflows. The figures accompanying the applications in this paper are generated using Fatbox tutorials and open-source data and results, ensuring reproducibility and providing users with easy access to the workflow. The toolbox is further supplemented on the GitHub repository with a glossary and a comprehensive documentation.





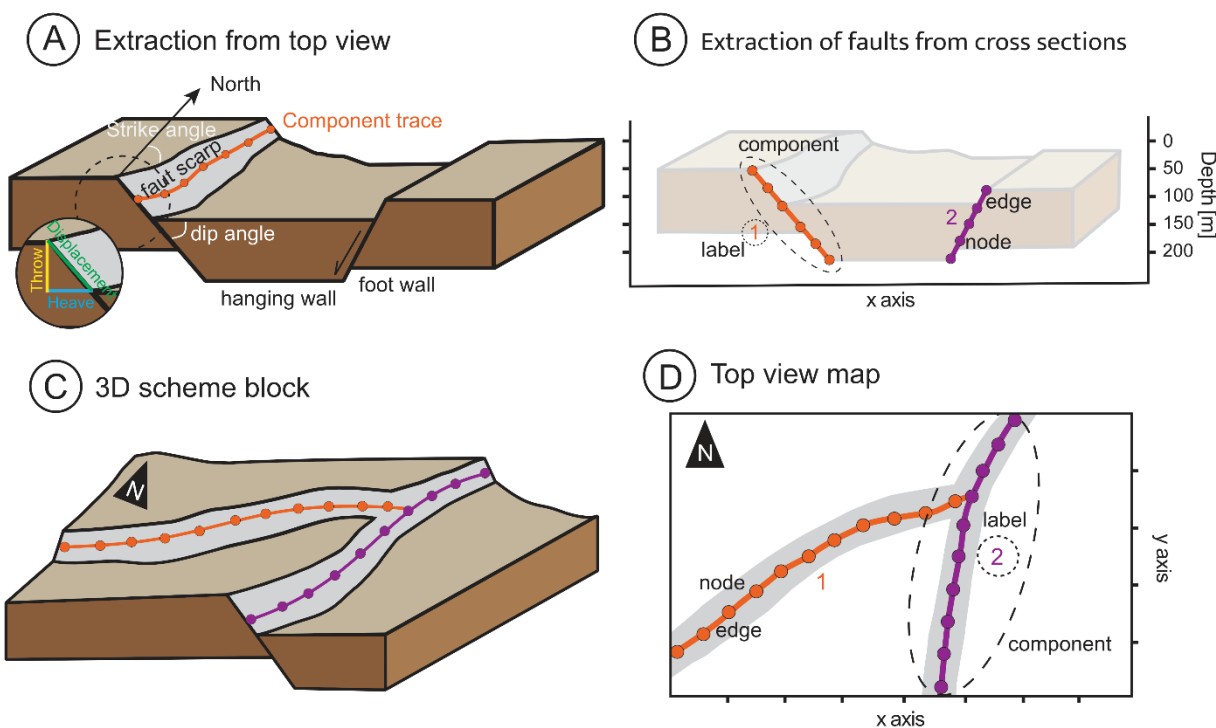

**Figure 1: Fault network description schemes.** The fault systems are described as networks, consisting of sets of nodes (depicted as circles in the figure) and edges (the segments connecting the nodes). The nodes are defined by their spatial (x, y) coordinates in 2D space, and the edges are defined as a pair of connected nodes. Components are groups of linked nodes; they represent individual faults. Components are often labelled via consecutive, integer fault identifiers. (A): Map view showing the location of the mapped faults on the scarp. Each fault is characterized by key geometric parameters: throw, heave, and displacement. (B) Fault traces on a vertical cross section. (C) The triple junctions (or y-nodes) are split based on their orientation, ensuring that the two closest branches remain connected while the splay is cut off. (D) Top view of triple junctions illustrating the geometric configuration.

## 2.2. Accessibility and structure of the toolbox

The Fatbox repository is available on Github https://github.com/PaulineGayrin/Fatbox

Fatbox functions are grouped in 6 different Python scripts that follow a typical sequential workflow.

The 6 scripts of the library are accessible on Github in the folder */modules*.

1. *preprocessing.py* - Prepare the dataset for fault network extraction.

2. *edits.py* - Extract the fault network from the dataset and edit the network and its sub-networks.

3. *metrics.py* - Compute various metrics of the fault network, such as length of the edges, node properties, number of components.

4. *plots.py* - Visualize the fault network and results of the analysis.



5. *utils.py* - Various low-level helper functions.

6. *structural_analysis.py* - Measure the geometric properties of the faults.

The data and workflows demonstrated in this study are available in the folder */tutorials*. The tutorials demonstrate the
possibilities and explain the steps of the process.

## 3. Applications

In order to demonstrate the capabilities of Fatbox, we present three different applications addressing typical challenges: (1) fault extraction and geometric analysis from topographic data of the Magadi-Natron basin, Kenya; (2) fault extraction and temporal correlation from strain data in a numerical rift model; (3) fault extraction from topography and particle image
velocimetry, geometric evolution tracking in an analogue rift model.

### 3.1. Application 1: Topography in the Magadi-Natron basin, Kenya

Digital Elevation Models (DEMs) represent normal faults at the surface as abrupt changes in altitude (Muirhead et al., 2016). Using remote sensing, elevation, vegetation and anthropogenic features can be depicted, and result in digital surface models, which we will refer to as DEMs for convenience. In geological studies, DEMs are a valuable tool for analysing the structural
framework and kinematics of tectonically active regions (Panza et al., 2024). For fault network analysis, the broad range of DEMs resolutions enables detailed investigations of small areas (tens of kilometres) with a high density of detected faults, as well as regional-scale studies (hundreds of kilometres) where only major faults are identified. In 2019, the European Space Agency released GLO-30, an open-source global DEM with a 30-meter resolution (Copernicus ESA https://doi.org/10.5270/ESA-c5d3d65). GLO30 provides a high-quality landscape representation based on commercial data
from TanDEM-X SAR satellites (Purinton and Bookhagen, 2021). Accordingly, our study utilizes GLO-30 to leverage freely available data while maintaining high-resolution quality.

### 3.1.1. The Magadi Natron basin

The East African Rift System (EARS) is the largest active rift on Earth and exhibits tens of thousands of normal faults that displace the surface sufficiently to be recognized in satellite-derived digital elevation models (Fig. 2). In the following
example, we apply Fatbox to the Magadi-Natron basin, which is located in the southernmost part of the Eastern Branch of the EARS in Kenya and Tanzania. The borders of the basin are limited by very mature normal faults while the intra-rift present partly more recent topography as the entire Southern Kenya Rift, were buried beneath a layer of Magadi trachyte approximately 1.2 million years ago (Baker and Wohlenberg, 1971; Muirhead et al., 2016). The trachyte lava flows have reset most of the basin's topography, meaning that the normal fault scarps in the Magadi-Natron Basin are no older than 1.2
Ma although earlier crustal weakness may have played a role in later fault localization. In addition, the fault scarps are well



preserved (Riedl et al., 2020) thanks to the aridity, sparse vegetation and minimal anthropogenic presence in the area. The basin features a dense and complex fault network with minimal cross-cutting structures, making it an ideal case study. To illustrate our workflow, we apply Fatbox to a 175 km² area covered by the DEM, accurately representing the natural complexity of the region.


**Figure 2: Context map of the DEM application.** The Magadi Natron Basin (B) is located south of the magma-rich eastern branch of the East African Rift System. (A) the major active faults are drawn in brown, from Styron & Pagani 2020, on © OpenStreetMap contributors 2025. Distributed under the Open Data Commons Open Database License (ODbL) v1.0. (B) the colours represent the topography using



GLO30 DEM. The blue shaded background represents the clipping mask used to focus on the intra-rift faults. The black dashed frame is the small area used in Fig. 3 and in the DEM tutorial.

### 3.1.2. Fault extraction from topography

We illustrate the extraction of fault networks from topography using the DEM GLO30 of the Magadi-Natron basin as input (Fig 3 A). First, a Gaussian blurring filter is applied on the topography to smooth small-scale ground variations that do not
constitute significant structural signals. This approach     enhances the signal-to-noise ratio of fault scarps. The edges of the scarp are then detected using the Canny edge detection algorithm (Canny, 1986), efficiently identifying edges of varying size, orientation, and shape. However, this method also highlights other topographic features, such as river boundaries, volcanoes, and lava flows. These non-fault-related components exhibit distinct geometric signatures; they are subsequently filtered out in using dedicated functions based on curvature and length. Next, the edges of the detected structures are reduced
to one-pixel wide lines through skeletonization using the thinning algorithm of Guo and Hall (Guo and Hall, 1992) (Fig 3 B). Other types of skeletonization techniques are described in section 3.2.2 and can be used per user decision. Finally, a connection detection function identifies adjacent pixels as connected components, assigning the same component identifier to pixels belonging to a continuous fault trace. Each pixel within the detected structures is transformed into a node, while the edges of the network establish connections between these nodes (Fig 1 D), effectively completing the network extraction
process (Fig 3 C). To avoid too dense node densities that impede performance, a filtering function is applied to selectively remove nodes at regular intervals. This approach optimizes computational efficiency while preserving a reasonable resolution.

As an alternative option to automated fault detection, a fault network can also be imported from previous manual mapping.
This allows the user to assess the quality of the semi-automated mapping against established and trusted datasets. Manually mapped faults are directly imported as geotiff. The fault traces are categorized as single structures in the connection detection step and the raster is then transformed into a network, analogous to the automated mapping process.





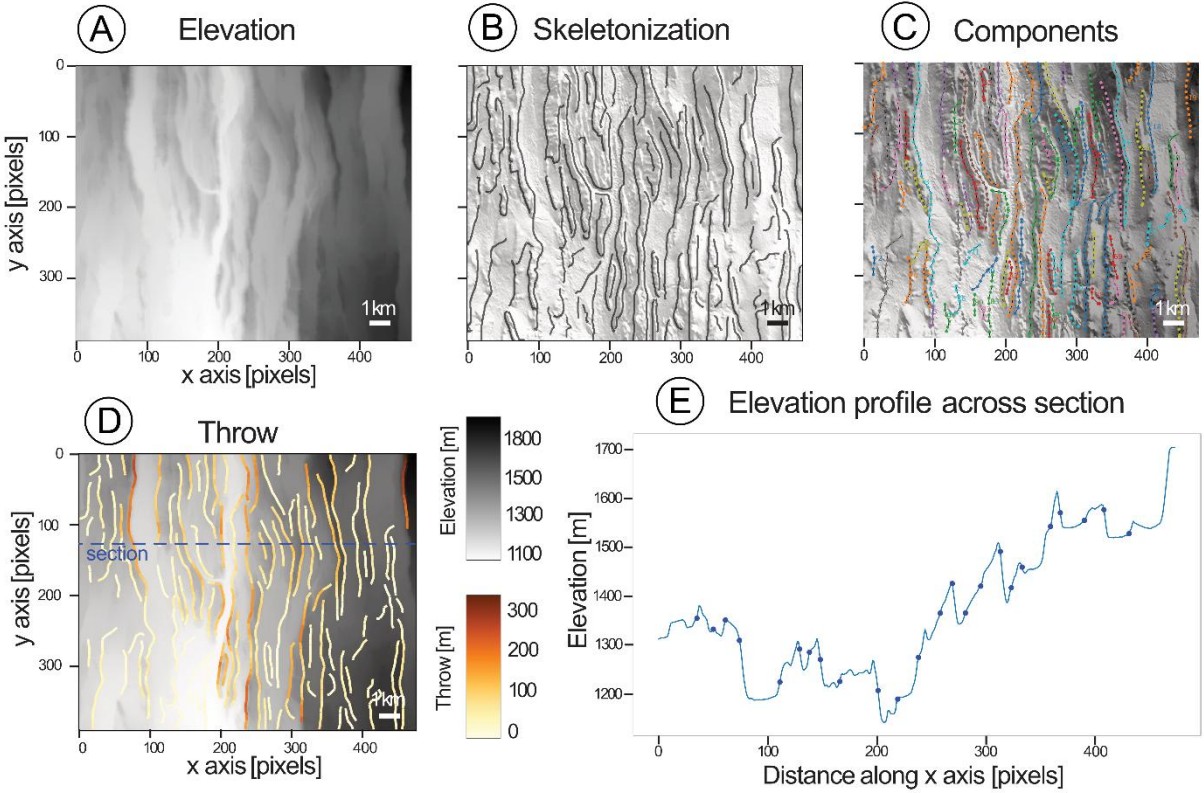

**Figure 3: Workflow of the DEM processing.** (A) Raw Digital Elevation Model, (B) Skeleton, detected structures are 1 pixel wide, plotted over a hillshade of the DEM. (C) Fault network over DEM and hillshade of DEM. Different colours represent different components. (D) Throw over DEM. (E) E-W cross section perpendicular to the main fault axis at latitude 1.37 S, location shown in panel D. The dots represent the position of the mapped faults.

### 3.1.3. Structural analysis

The surface expression of faults contains a wealth of structural information, which is critical for geological interpretation. Through fault extraction, the centre of the fault scarp along each fault trace is accurately determined (Fig. 1A). Next, Fatbox performs a systematic and automated structural analysis for every edge along the faults, emulating the detailed procedures typically carried out during fieldwork. Diverse geometrical attributes are measured (Fig. 1A). The length of a fault is computed by summing the lengths of the edges belonging to the component, extending from one fault tip to the other. The azimuth of the fault (strike angle), is measured from top-view perspectives, with respect to the North. To determine the scarp geometry, a virtual cross section is drawn perpendicular to the fault axis until reaching a predefined maximum distance, the maximum vertical offset between the hanging wall and footwall is interpreted as the fault throw. Fault displacement and extension are subsequently derived, assuming a dip angle of 60°. All geometric attributes are systematically stored, whereas



each component of the fault network acts as a python dictionary-like data structure. Geometric information pertaining to an entire component is accessible through dedicated attributes. Additionally, each node and edge within the component

possesses its own set of attributes: nodes are characterized by properties such as spatial position and the number of neighbouring connections, while edges include attributes such as length, the nodes they connect, strike, and extension.

Although the Magadi-Natron Basin is a relatively arid region, erosion remains a significant factor affecting fault scarp morphology. Erosional processes remove material from the footwall, rounding the initially sharp fault scarp, while

deposition occurs onto the hanging wall, partially infilling the bottom of the fault surface. To mitigate these effects, throw measurements are taken at a reasonable distance from the fault crest, and a 60° fault dip is assumed for extension calculations (Fossen, 2016; Riedl et al., 2022). This dip angle is consistent with previous displacement-length studies in the Kenya Rift (e.g., 60° in (Muirhead et al., 2016); 65° in (Shmela et al., 2021)). Additionally, apparent fault dip measurements can provide insights into the extent of erosion in a given area. As all the geometric attributes are stored for every edge,

statistical analyses can be conducted on groups of faults based on criteria such as azimuth orientation and displacement. This facilitates comparative studies between faults or among different fault families. Furthermore, integrating the fault network with digitized geological maps enables comparisons between faults of similar maximum ages. Ultimately, structural analysis provides a comprehensive and quantitative characterization of fault surface expressions of the area.

**3.2. Application 2: Numerical modelling**

Numerical forward models have become a key approach in geodynamics where they are used to study a variety of processes across a broad range of scales, from large-scale mantle convection (Coltice et al., 2018) to subduction (Pons et al., 2022), rifting (Jourdon et al., 2020) and strike-slip tectonics (Heckenbach et al., 2024). Recent advances in computational techniques have allowed high resolution 2D simulations yielding new insights into rift migration processes (Brune et al.,

2014), deformation phases (Naliboff et al., 2017), and fault-related unconformities (Pérez-Gussinyé et al., 2020). In continental rift systems, crustal strain localizes in zones of brittle deformation, gradually weakening the surrounding rock. As this weakening progresses and strain continues to build, the rock eventually reaches a point of mechanical failure, leading to the formation of faults. Numerical models simulate the evolution of strain and strain rate over time, enabling the identification of fault formation and the tracking of faults when a predefined mechanical threshold is surpassed. However,

while numerical models provide valuable insights into the long-term evolution of fault systems, the conducted fault analysis is often limited to discrete time snapshots for direct comparison with natural data or is performed qualitatively. Additionally, because faults in these continuum models are represented as finite-width brittle shear zones, extracting slip rates and other quantitative parameters characteristic of discrete natural faults remains challenging. Recent studies on the quantification of evolving fault network characteristics in 3D simulations of continental rifting (e.g., (Duclaux et al., 2020; Naliboff et al.,

2020; Pan et al., 2022, 2023) have developed techniques to address these limitations. Fatbox builds on this work by offering





a framework to quantitatively extract and track the structure and evolution of faults through time, facilitating more accurate comparisons between numerical models and natural geological systems.

### 3.2.1. Numerical model setup

To illustrate the functionality of tracking temporal fault evolution, we use a 2D continental rifting model (Neuharth et al., 2022) where the geodynamic code ASPECT (Gassmöller et al., 2018; Glerum et al., 2018) has been coupled to the landscape evolution code FastScape (Braun and Willett, 2013; Yuan et al., 2019). This model simulates a continental rift while incorporating sedimentation and erosion processes. The model domain is 450 km wide and 200 km deep. Prescribed boundary conditions enforce 5 mm/yr extensional velocities on each side, resulting in a total extension rate of 10 mm/yr, while inflow is prescribed at the bottom boundary to maintain volume conservation. The 120 km thick lithosphere consists of 20 km of wet quartzite upper crust, 15 km of wet anorthite lower crust and 95 km of dry olivine mantle lithosphere (Fig 4 A). Beneath the lithosphere lies a weak asthenospheric layer composed of wet olivine (Neuharth et al., 2022). Rifting is initiated by thickening the upper crust to 25 km at the model's centre, leading to an initially warm and weak rift centre that ultimately evolves into a symmetric rift. Fatbox is employed to extract the faults and track their geometric parameters throughout the entire rift evolution.

### 3.2.2. Fault extraction from strain data

In these numerical models, faults are interpreted as zones of high strain or strain rate (Fig 4 A). Here we employ a strain-based fault criterion that distinguishes areas belonging to faults via a user-defined threshold. The thresholding assigns a value of 1 to pixels representing faults (where strain exceeds the threshold) and 0 to background pixels (where strain is below the threshold). The choice of thresholding method depends on the dataset: (1) an absolute threshold, (2) a relative threshold (e.g. 90% of the maximum strain of the model) and (3) an adaptive threshold (e.g. 90% of the maximum strain of a model in a moving window). In the following, we use a relative threshold optimized to highlight fault structures. We then apply a thinning algorithm to extract the skeleton of each shape, reducing fault zones to single-pixel-wide lines. Various skeletonization algorithms exist, differing in quality and speed (see the review of Saha et al., 2016). Here we use the efficient Guo and Hall algorithm (Guo and Hall, 1992). Once skeletonized, individual faults can be distinguished and classified separately by identifying neighbouring pixels as part of the same connected component (Fig. 1B). Finally, the fault network is generated from the labelled components by converting each pixel to a point and connecting neighbouring points (Fig. 4 B). In some cases, interpolation to a regular grid is necessary, particularly when the model dataset has variable resolution (as in Neuharth et al., 2022). The final output is a structured network of nodes and edges (see Fig. 1 B), where each fault is identified with a unique label. The extracted network may require additional filtering to remove computational artefacts or geometric complexities introduced during processing. Our library provides functions to address these issues. In particular, the raw extracted network is highly dense, with a node at each pixel belonging to a fault and edges connecting them. This excessive density can lead to unnecessarily long computation times. To optimize the network, a filtering function removes





selected internal nodes (based on user-defined parameters), reducing density while preserving realistic geometric orientations.

### 3.2.3. Time stepping, correlation, and structural analysis

The fault extraction is performed independently at each time step, meaning that structures receive labels that may vary between steps (Fig 4 B - C two consecutive timesteps). In order to track the evolution of the network, however, each fault must retain a consistent label throughout the workflow. We therefore introduce a correlation step that ensures that the same fault is identified by the same label across all time steps. This process is complicated by the fact that faults can advect, grow, shrink, appear, disappear, merge, or split. To account for these changes, the program compares each fault at time 1 with each fault at time 2 and vice versa i.e., forward and backward. The connections are established by computing the similarity between faults of different time steps (Fig. 5). Specifically, we calculate the minimum distance between each node of a fault and each node of the neighbours and then average these distances. The faults are considered similar and correlated when the average distance between them is lower than the set distance R (Fig 5). This method allows users to loosen or tighten the correlation by varying R. Once correlations are established, faults from the second time step are relabelled according to the corresponding faults from the first-time step, ensuring continuity. The program effectively tracks fault evolution over time by assigning each structure the label it originally received in the first-time step, maintaining a historical record.

Once faults are correlated through time, the temporal evolution of the fault system can be analysed. The article (Neuharth et al., 2022) provide a detailed study of fault system evolution in the 2D numerical models discussed earlier, coupling continental rifting with surface processes. The structural analysis is performed at every time step and the geometric data are stored for the entire fault network. This dataset enables the tracking of kinematic properties such as displacement over time, allowing for the study of individual fault growth as well as the development of the entire fault system.



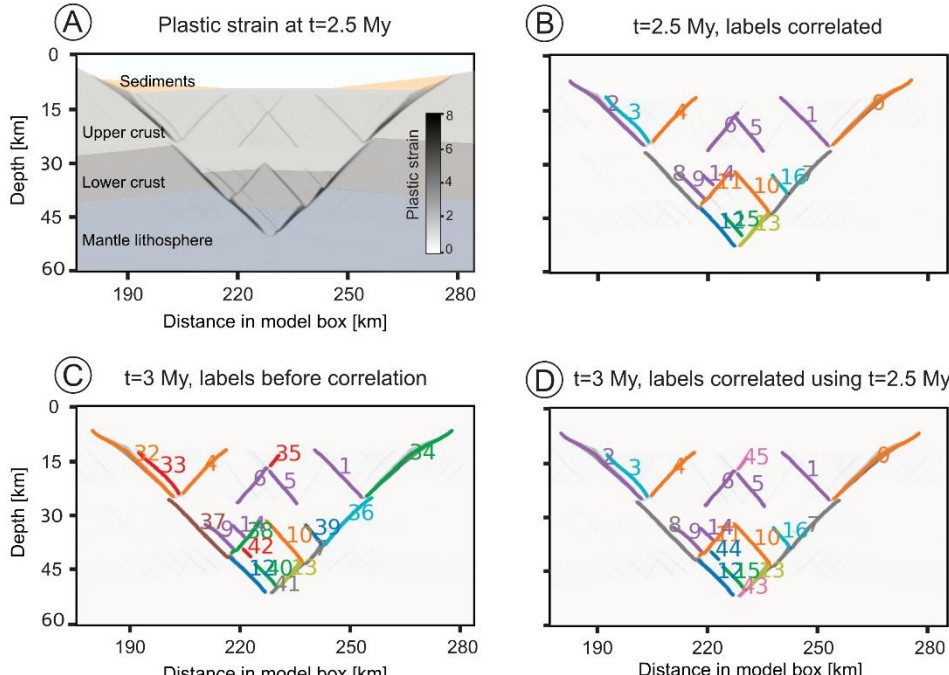

**Figure 4: Fault correlation in the numerical rift model.** A) Plastic strain at t=2.5 My, the network is mainly developed at the centre. B) Fault network extracted at t=2.5 My correlated using earlier and later time steps (forward + backward correlation) C) Fault network extracted at t=3 My (before correlation). D) Fault network at t=3 My, correlated using labels at 2.5 My. In practice, the resolution between timesteps may be much higher. This interval was chosen here to exhibit enough difference to show the effect of the correlation. The labels of the faults are shown as numbers next to the corresponding faults using the same colour.





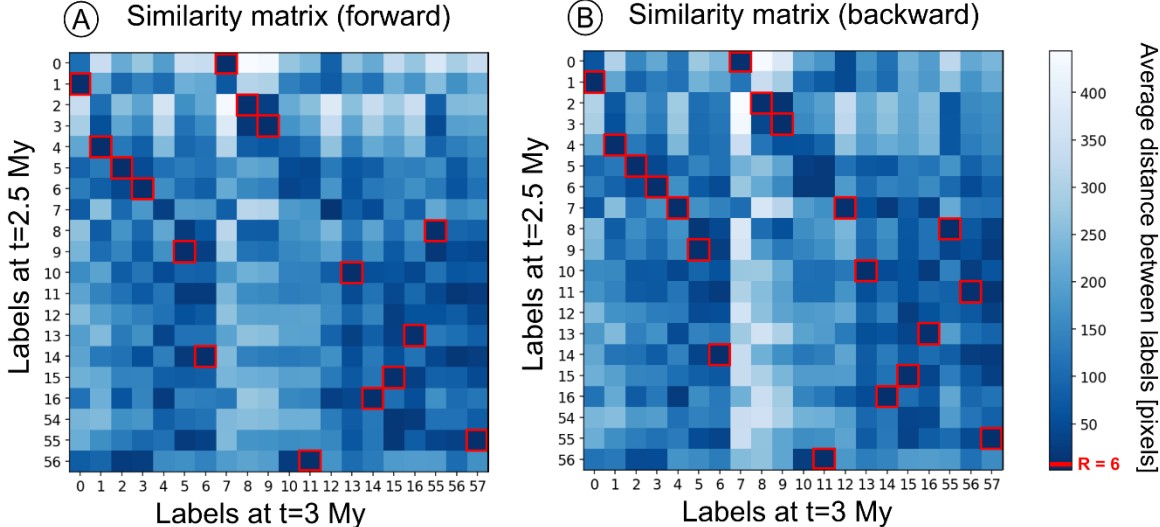

**Figure 5: Correlation of faults across two time-steps.** The similarity matrix shows all the labels of the pair of graphs correlated. In A, the forward direction means the graph at t=2.5 My is used to correlate the graph at t=3 My. In backward direction (B), the base for the correlation is the graph at t=3 My. The red square indicates the labels distance below R the correlation threshold. They have the lowest distance, thus the highest similarity.

### 3.3. Application 3: Analogue modelling

Analogue modelling is a widely used technique in geosciences for replicating natural scenarios in a laboratory setting under controlled conditions with appropriately scaled lengths, times, and forces. The practical workflow of analogue modelling typically consists of model setup, preparation, and experiment execution, which is usually monitored by digital cameras. The cameras comprise a monitoring system which is able to measure surface strain and/or elevation from sequential images of the laboratory experiments (Fig. 6). The most common technique, widely known as Particle Image Velocimetry (PIV) or Digital Image Correlation (DIC), calculates velocity fields by cross-correlating digital images, detecting texture patterns on the surface, sides, or interior of the model (Strak and Schellart, 2016). Topographic data is derived from these digital images using photogrammetry techniques. When photogrammetry is unavailable, alternative methods such as laser scanning (Willingshofer and Sokoutis, 2009) or laser interferometry (Strak et al., 2011) can be employed to obtain elevation measurements.

Fault extraction is typically performed manually by most analogue modelers (Philippon et al., 2015; Schlagenhauf et al., 2008), a process that is both very time-consuming and tedious. In this application, we reanalyse a study on rift propagation in a brittle-ductile multilayer analogue model (Molnar et al., 2017). Fatbox is integrated into standard methodology that



combines time-tracking of faults and structural analysis of the topography. Two representative workflows for semi-automated fault extraction and analysis are presented, employing elevation and strain data obtained from the experimental setup.

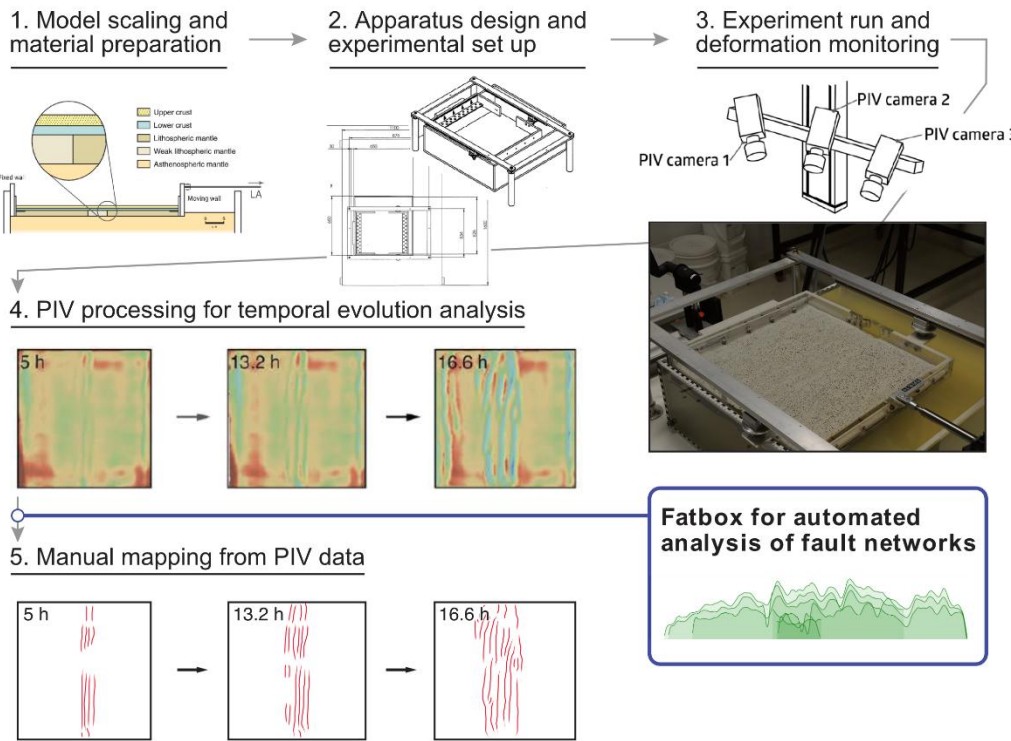


**Figure 6: Integration of Fatbox in the analogue modelling workflow.** The program replaces manual mapping and automatically analyses the fault geometry and network evolution.

### 3.3.2. Fault extraction from elevation data

The digital photographs are captured at predefined time intervals, and the surface topography data is subsequently calculated.
The fault extraction workflow is similar to the processing of DEM data (see section 3.1.2) from satellite imagery. First, a Gaussian blur filter is applied to mitigate noise with a carefully chosen smoothing window, preserving the integrity of structural features. The background noise of analogue models is usually low and requires minimal smoothing. Feature edges are then detected using the Canny algorithm (Canny, 1986), which identifies intensity gradients in the images—serving as proxies for fault scarps. To transform these detected edges into lines, skeletonization is performed, refining the extracted
lines to a one-pixel width. A connection detection function then identifies adjacent pixels as connected components, assigning the same component ID to pixels belonging to a continuous fault trace. Each pixel within the detected structures is



transformed into a node, and the edges are generated to connect nodes based on their proximity, allowing faults to be clustered and labelled. Since this process may also detect non-fault features or artefacts, a final cleaning step is applied, considering attributes such as fault length, shape, and curvature. These operations are performed across all time increments
in the dataset. Finally, all time steps are correlated to maintain the consistency of the labels (see 3.2.3).

### 3.3.3. Fault extraction from incremental strain data

Surface strain is computed by comparing one set of images with the subsequent set, providing values for incremental strain. Although this example applies the workflow to top-view images, it can also be adapted for side-view images of laboratory
experiments. Faults are identified similarly to numerical models using a strain threshold to distinguish active faults. They are highlighted relative to the background using an absolute threshold. The subsequent workflow is the same as for the numerical model: data are binarized (1 for fault and 0 for background) and skeletonized using the Guo and Hall algorithm (Guo and Hall, 1992). The connections are detected with each pixel being assigned the same component ID as its neighbouring pixels. The pixels are then converted into a node, and the collection of connection edges for each time
increment is transformed into a network. Depending on the pre-processing stage (PIV strain computation and data export), incremental strain datasets may contain varying levels of noise, heterogeneities, or areas with missing values. These issues can result in the incorrect identification of artefacts as faults. An additional cleaning step is introduced to address this problem, which includes: (1) removing nodes connected to three or more other nodes, (2) slicing fault traces into smaller segments by eliminating nodes where the direction of the line edges deviates significantly (<120°) from neighbouring edges,
and (3) removing components smaller than a defined length threshold. After cleaning the network, the final step involves time-stepping and correlating faults across all time increments in the experiment dataset (Fig 7 A). When both incremental strain and elevation data are available, it is advisable to perform fault extraction on both datasets, compare the results, and use the one that yields better outcomes based on the available data.

### 3.3.4. Structural analysis

The structural analysis is conducted as outlined in 3.1.3 for each time step. In this case the measured dip can be directly used to compute the displacement and extension of the faults as erosion is absent. The correlation and proper labelling of faults between consecutive time steps enable the visualization of the evolution of specific geometric parameters over time. For example, the throw as a function of fault length (Fig. 7 B) is commonly used to enhance the understanding of normal fault growth (Lathrop et al., 2022; Schlagenhauf et al., 2008) and to assess the growth stage of the fault network.




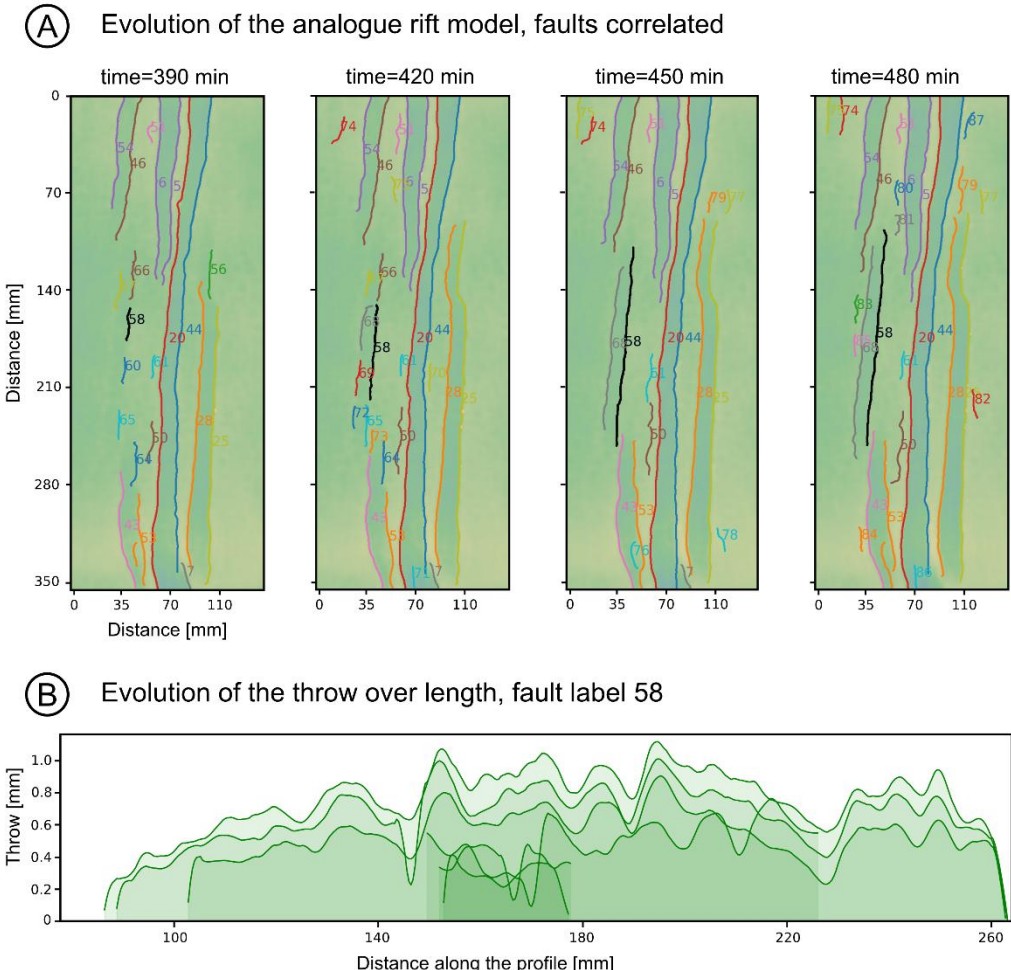

**Figure 7: Time tracking on analogue data.** A) Evolution of the network as the extension increases in the analogue model. The faults are correlated and tracked through time. B) Throw length profile of the fault label 58. This ID is highlighted on A in black.

## 4. Discussion

### 4.1 Fatbox options for defining a fault

Fault extraction requires a well-defined identification criterion, tailored to the specific dataset and the desired level of detail. This paragraph compares the definition employed in each dataset. In the case of the numerical model faults are defined as zones of locally high plastic strain. In analog models DEM are usually captured at set time steps, where the resolution has to be sufficient to capture the evolution of the network. In the DEM analysis, normal faults are characterized by a sharp change in elevation along a relatively linear trajectory, with a typical evolution (Figure 7B) of displacement along the fault. The



Canny edge detection is based on this topography gradient criterion. Other structures, such as volcanoes, riverbeds, and orogens, can also cause sharp changes in topography in the field, but they are usually well-mapped, well-known, and exhibit specific shapes. The network description used in the toolbox allows to capture the complex geometry that can appear during the fault systems evolution (Claringbould et al., 2020; Henza et al., 2010). As faults are described as a set of interconnected nodes, a large degree of geometric freedom is enabled where neither the nodes nor the connections are constrained in any way.

The toolbox provides users with the flexibility to adapt the fault network model to the specific requirements of diverse geological scenarios. For example, in complex configurations such as triple junctions—where three fault branches connect— users can choose to either preserve the junction by treating the splay as part of a continuous fault, or to split the junction by defining the splay as a separate fault segment. These decisions are particularly significant in mature fault systems, which often exhibit a high degree of structural complexity due to multiple phases of deformation, resulting in numerous splays, junctions, and intersections.

## 4.2. Why use Fatbox?

The examples presented in this study illustrate the versatility of Fatbox in mapping fault systems across a range of settings. The toolbox provides an extensive suite of functions that facilitate the construction of customized workflows for the extraction and analysis of fault networks. These functions are designed to be fully interoperable, enabling users to tailor workflows to the specific characteristics of their data and the geological context under investigation. Furthermore, each step of the process can be executed independently, offering a high degree of flexibility. Manually mapped fault networks for instance can be imported in the early steps of the workflow to use the structural analysis to compute along-strike fault displacement. Compared to traditional manual fault mapping approaches, Fatbox presents several key advantages, including increased efficiency, reproducibility, and adaptability to complex fault geometries. Semi-automated fault mapping, as implemented in Fatbox, offers a transparent and efficient alternative to traditional manual approaches. It relies on image processing techniques, open-source code, and a minimal number of user-defined parameters. Workflows can be adapted according to user constraints due to the modularity of the functions.

Unlike manual fault mapping—which is inherently subjective and difficult to reproduce due to the undocumented nature of interpretive decisions—semi-automated methods reduce interpreter bias and promote consistent results. Fatbox distinguishes itself from existing workflows, which often focus on isolated tasks such as fault mapping (e.g., Dirnberger et al., 2015; Mattéo et al., 2021; Nyberg et al., 2018) or structural analysis (e.g., Stewart et al., 2018; T et al., 2025). In contrast, Fatbox integrates both functionalities within a single framework, and further extends its capabilities by enabling temporal correlation of fault identifiers, using only image processing techniques. The computational requirements remain modest: most steps can be parallelized, resulting in short runtimes even on standard desktop hardware. For instance, the complete





fault extraction process for the numerical rift model was executed within a few hours, demonstrating the practicality of the approach without the need for resource-intensive methods.

## 4.3. What are the limitations?

Fatbox represents a promising approach to fault network analysis, but its current implementation leaves some challenges unaddressed. Primarily, Fatbox is optimized for the extraction and analysis of fault and fracture networks in 2D, yet faults and fractures are inherently three-dimensional structures. Although two-dimensional representations—such as surface maps and cross-sections—are commonly employed due to observational constraints or the relative geometric consistency of certain structures along strike or dip, they cannot fully capture the complexity of the three-dimensional nature of fault
systems. Consequently, further development is required to enable the incorporation of three-dimensional data into the fault characterization process (e.g. Jourdon et al., 2025).

Another limitation is the introduction of artefacts during the fault extraction, which arise from background noise or from the presence of non-fault structures. While many artefacts possess distinct geometric or textural features that allow for their
removal following initial network extraction, some closely resemble true fault structures and are therefore more challenging to identify and exclude. For example, in the analysis of intra-rift topographic data, features such as lava flow boundaries or river channels may be inadvertently included in the fault network due to their similar morphologies. Although such features may differ from faults in aspects like curvature (La Rosa et al., 2025), distinguishing them with high confidence remains difficult. To mitigate the influence of artefacts, we recommend that users apply data preprocessing techniques—such as
smoothing, masking, or band filters—to minimize the artefacts presence prior to extraction. Additionally, it is advisable to conduct a post-extraction validation of the identified fault network against established geological observations, at least on a representative subset of the data, to benchmark the reliability of the semi-automated extraction process and adjust mapping settings.

In addition, the resolution of the extracted fault network is inherently constrained by the resolution of the input dataset. This dependency can be strategically utilized to enable rapid mapping of extensive areas using medium-resolution data, with the possibility of employing higher-resolution datasets for more detailed analyses. However, while higher-resolution inputs can enhance spatial detail, they also tend to increase the likelihood of introducing artefacts. As such, a careful balance must be found between resolution and interpretive accuracy. Users are therefore encouraged to select data resolutions that align with
the scale and objectives of their analysis, taking into account both the desired level of detail and the potential for noise-induced artefacts.



### 4.4. Outlook

The current limitations of the Fatbox toolbox point to the potential for future development. While the toolbox can be applied to a series of 2D slices or cross-sections to infer the 3D geometry of fault systems, a complete 3D implementation remains a
critical next step. Many of Fatbox's existing functions could be extended to operate within a 3D framework; however, this transition introduces new challenges. In particular, defining fault network connectivity in three dimensions is non-trivial, as fault splays, junctions, and intersections become substantially more complex with the added degree of freedom. Encouraging progress has already been made in this direction. For instance, (Jourdon et al., 2025) demonstrated an approach in which fault traces extracted via medial axis transform of a 3D forward numerical model were interpolated into a 3D point cloud
using the Delaunay algorithm. A similar methodology could be incorporated into Fatbox to facilitate the reconstruction of 3D fault surfaces. In parallel, additional geometric and kinematic analyses—such as the conversion of fault throw to strain (La Rosa et al., 2025)—could be integrated to broaden the analytical capabilities of the toolbox. These enhancements are planned for inclusion in updates of Fatbox.

Finally, we encourage contributions and collaborations from across the geoscientific community. Whether applied to the analysis of plate boundaries in global numerical models or to the interpretation of fault systems in bathymetric or seismic data, Fatbox offers a flexible framework with broad potential. Community-driven extensions can be incorporated into the toolbox via GitHub, helping to establish Fatbox as a shared and evolving resource for fault network analysis.

### 5. Conclusions

Fatbox provides a versatile and comprehensive suite of tools for the extraction and analysis of fault networks across a range of data types. These include topographic datasets such as digital elevation models and analogue model outputs, as well as strain fields derived from numerical simulations or particle image velocimetry. Fatbox combines methods from computer vision and network analysis, facilitating the semi-automated geometric and kinematic analysis of fault systems. This integration significantly accelerates workflows, allowing for efficient and reproducible fault network characterization. A key
strength of the toolbox lies in its ability to treat fault networks as dynamic, evolving systems, enabling users to track changes in structural parameters over time. The toolbox also supports the direct quantification of fault geometry from topographic surfaces, enhancing its utility for geomorphological and structural analyses. By enabling quantitative fault analysis, Fatbox serves as a valuable resource for researchers aiming to investigate the development and evolution of fault systems across spatial and temporal scales.






**Author contribution**

PG, TW: conceptualization, methodology, software, formal analysis. PG, TW, NM, DN, SB: investigation, visualization. SB: supervision, funding acquisition. PG, TW, NM, ALR, DN, JN: manuscript draft preparation.

**Competing interests**

The authors declare that they have no conflict of interest.

**Financial support**: P.G. has been funded by the German Science Foundation (DFG) (Project No. 460760884). S.B. acknowledges funding from the European Union (ERC, EMERGE, 101087245).

**Acknowledgments**

The authors thank Tim Hake, who contributed to a former version of the fault extraction workflow.

**Code availability**

The library Fatbox is available on Github https://github.com/PaulineGayrin/Fatbox . Contributions are welcome.

**Data Availability**

The 30 m resolution GLO30 DEM is publicly available via Copernicus ESA https://doi.org/10.5270/ESA-c5d3d65. The timesteps resulting of the numerical models and of the analogue model are available on the Fatbox Github repository and comes from published work ((Molnar et al., 2017; Neuharth et al., 2022). The figures in this study have been generated using Python3, QGIS and Affinity designer 2.

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
