# Peer review of "Fatbox: the fault analysis toolbox"

_EGUsphere, 2025_

## Author Comment (AC2)

I reviewed the paper entitled "Fatbox: the fault analysis toolbox" by Gayrin et al. The paper presents a python software using image analysis methods to automatically extract faults from DEMs, numerical and analogue models in two dimensions. In addition to reconstruct a fault network, the software can perform a time evolution analysis to characterize fault evolution in time and space. Although limited to 2D applications, as well discussed by the authors in the limitation section, the method looks good and the presented applications demonstrate well that it is promising.

Thank you for taking the time to review our paper and for your very constructive comments.

Here are some comments and remarks:

The paper presents the results of applying a sequence of methods and functions to treat the data and obtain a fault network. Although the results look good, I think it would be nice to provide more details about these functionalities and illustrate the effect of the functions applied at each step from the raw data to the final fault network. While some steps are shown for the DEMs fault extraction, we do not see it for the numerical model to reconstruct a fault network from the raw plastic strain output. The jupyter notebooks illustrate this more, but I think it would be good to have these steps in the paper.

Agreed. To illustrate the fault extraction steps, new panels have been added to figure 4 for the numerical model application, and to figure 7 for the analogue application.

You made the choice to structure the paper using applications which leads to a lot of redundancy as some methods are the same across the applications e.g., the DEM and analogue model share the same automatic topographic analysis while the numerical and analogue models share the same approach to use strain and characterize fault evolution over time and space. I suggest that instead of structuring the section 3 by application you could structure it by approach or functionality and present how a functionality is used for each application. At the beginning of section 3 you could present the applications, and then describe your functionalities, how they work, what they do, when to use them etc. with some nice illustrations.

The paper presents a tool that can be used by three communities who don't necessarily interact. We adopted this structure to allow readers to focus on the applications most relevant to them without missing essential information. This approach indeed introduces some redundancy, which we consider beneficial for clarity and accessibility.

There is a problem with the references across the paper. I don't know how this occurred but there are a lot of citations in the text that are not in the reference list. You should double check that before re-submitting the article.

Thanks for finding this issue. There was a glitch in the reference manager use. The missing references have been added.

Finally, because the paper is very oriented about the Python package, it is referred several times and its content is described, I had a closer look to it. I have some suggestions that I think could benefit its spread and use but I do not think that the authors need to address them for the paper to be published.

**Line by line comments:**

L107: "Each component of the network consists of nodes (points defined by location x- and y-coordinates) and edges (connections between nodes)"

This sentence seems appropriate to introduce here that this (nodes + edges) is called a graph, particularly because it is the title of the subsection and that you use the term graph at line 112. Manuscript modified as suggested by the reviewer.

L137: I am sorry, but I do not understand what "edit the network" means in this context. This sentence refers to the functions from the *edits.py* module that allows to modify the structure of the graph (example: removing nodes, removing triple junctions, label components) or compare graph between them (example: find common component). The manuscript has been modified to clarify this sentence.

L185-187: What is actually detected by the Canny algorithm? In the jupyter notebook we can see that the method is conveniently implemented in the scikit library but the actual quantity and approach utilized to detect "edges" and what it means in the physical world is not clear to me. It is a very important step of the method presented here, and even if it is not implemented by the authors, it would be nice to have an overview of the approach.

When applied to topography data, the "edges" align with strong changes of topography ie. slopes. Manuscript modified to clarify this step.

L188-189: "non-fault-related components exhibit distinct geometric signatures; they are subsequently filtered out in using dedicated functions based on curvature and length" Interesting! However, there are no real details in the manuscript on how this important step is taken. Looking into the jupyter notebook tutorial we can see that the scikit method called "remove\_small\_objects" is used, with a parameter "min\_size=30" pixels. This value sounds arbitrary. Is there a generic way to choose it? If the surface of the covered ground or the

resolution of the DEM was different, would that value change? L215: "a virtual cross section is drawn perpendicular to the fault axis until reaching a predefined maximum distance" How is that predefined maximum distance chosen? In the notebook it is set to d=12 (no units indicated so I assume it is also in pixels?), same question than in the previous comment: will that value change if the surface and/or resolution of the DEM changes? How can you be confident enough that using a constant value for all faults may not lead to crossing another fault depending on where you are performing the analysis?

These parameters are defined by the user to best reflect the characteristics of the study area and its geological context. A new Section 2.3 has been added to the manuscript to describe the parameter settings and provide guidance on selecting appropriate user-defined values.

L271: "In the following, we use a relative threshold optimized to highlight fault structures." Could you please provide a bit more details of how the optimized relative threshold is chosen? The optimum threshold is chosen by the user, depending on the level of detail wanted. Manuscript modified to detail this step.

L277: "In some cases, interpolation to a regular grid is necessary, particularly when the model dataset has variable resolution"

Why is that step necessary? If the object used in the first place is an image, the pixel grid is already regular even if the model's mesh is not, isn't it? Can you please provide an example for which that step is required and illustrate why?

In our example, we needed to interpolate to an evenly spaced grid because ASPECT uses adaptive mesh refinement with spatially variable resolution. To overcome this, we first load the variable grid from ASPECT and then interpolate this onto an evenly spaced grid of pixels before applying the next steps (e.g., thresholding, skeletonization). Manuscript modified to clarify this step.

L280: "Our library provides functions to address these issues"

This is interesting, but unfortunately, we miss details here. What are these functions? Could you be more specific about which function addresses which issue and how? It would help the readers to know what functions they should use depending on their case.

Manuscript modified to address this step. Examples have been added.

L282-284: "To optimize the network, a filtering function removes selected internal nodes (based on user-defined parameters), reducing density while preserving realistic geometric orientations." Could you please elaborate on that function? What does it actually do?

Manuscript updated to reflect this issue. During classification, the algorithm places a node in every pixel. This high density is usually not necessary, it only leads to a slower workflow. The filtering function removes a node every X nodes (user-chosen value), keeping the significant geometry while ensuring faster execution.

L290-291: "the program compares each fault at time 1 with each fault at time 2 and vice versa" If I understand correctly, you mean that you compare two consecutive timesteps, right? If this is the case, I suggest to use "time n" and "time n+1" instead of "1" and "2" to better emphasize that it is between two consecutive steps, whatever their numbering.

There are several levels of correlation. In the case of a level 1 correlation, the program compares each fault at time n with each fault at time n + 1 (forward) and vice versa: each fault from time n+1 with each fault from time n (backward). The level of the correlation is defined by the user. It is therefore possible to combine a correlation between time n and n + 1, with a correlation from time n and n + 2, and even more correlations. This choice is made according to the time resolution and the complexity of the network evolution. We explain this better in the revised manuscript.

L292-293: "we calculate the minimum distance between each node of a fault and each node of the neighbours and then average these distances"

I do not understand what is the "distance of each node of the neighbours". Neighbour is not defined and thus ambiguous. Could you please be more precise about the distance between what and what is computed?

The correlation step looks at every node of a component and computes the distance between each of them and all the other nodes of the graph at a given timestep, and then adds the computed distances to a similarity matrix. The comparison between the average distances of two faults defines their global similarity. Manuscript modified to clarify this step.

L299-300: "The article (Neuharth et al., 2022) provide a detailed study of fault system evolution in the 2D numerical models discussed earlier"

I think that it is a bit unfortunate that the illustration of the method presented in this paper is actually not in this paper but in another. You could maybe showcase some of the possibilities of the method presented here in a figure to provide the readers with examples of what they could expect from your software especially what you describe lines 302-303

Agreed. The figure 4 has been modified to illustrate the structural analysis performed using the numerical model.

L339-350: Section 3.3.2 roughly contains the same information than section 3.1.2. I suggest that you could remove this section.

As this paper targets three different communities, some redundancy is desired to make every application section understandable and stand-alone. However, we will consider reducing the level of detail in Section 3.3.2 and instead referring readers to Section 3.1.2 for overlapping material.

L355: "strain threshold to distinguish active faults"

Is strain correctly employed here? It sounds strange to use strain to identify active faults. Strainrate would be a better quantity because strain also contains inactive faults.

We agree, the sentence was misleading. Strain is used to identify active and inactive faults alike, while strain rate allows active faults detection. The manuscript has been corrected to clarify this point.

L357: "data are binarized"

I suggest to introduce the term binarized in section 3.2.2 where the procedure is first described. Manuscript modified to clarify the vocabulary as suggested by the reviewer.

L356-360: This paragraph is also a redundant information with a previous section. I suggest to remove it.

Similar to the DEM extraction paragraph, some redundancy is desired to make every application section understandable and stand-alone.

L362-365: Is this procedure different than the one described at lines 195-197? If not, I suggest to group them, if yes, I suggest to provide more details at lines 195-197 and then explain why here the procedure is different.

The employed function is the same. The manuscript has been modified according to the reviewer suggestion.

L387-388: "The Canny edge detection is based on this topography gradient criterion." While the fact that the Canny edge detection uses topography gradient can be reminded in the discussion, I think it would be nice to introduce much earlier, at lines 185-187, what the Canny algorithm actually does and what type of quantity it uses. For example, the use of topography gradient criterion was not mentioned before the discussion.

Agreed. We better defined what the Canny algorithm detects when it's first introduced.

**Section 4.1 Fatbox options for defining a fault:**

I feel like there could be more information about how to choose some parameters for each identification criterion. While the DEM analysis is slightly more detailed, the strain or strain-rate approach is summarized in a single sentence. The last paragraph is good, it states what type of choices can be made. Aren't there more like this that you could elaborate upon to provide more insights about your toolbox and what we could do with it?

Agreed. The manuscript has been modified according to the reviewer suggestion.

L420: "most steps can be parallelized"

What do you mean by "parallelized" in that context? Does it mean that your toolbox should run in parallel i.e., multiple mpi ranks/threads or that the steps of the procedure are independent of each other and thus you can perform each of them independently?

Because as I understood, there are steps or procedures that do not seem independent, like the time evolution and the correlation between time steps as each time step needs to be treated

**sequentially.**

Parallelization can be achieved by running the code on multiple CPU cores. While fault extraction is inherently sequential—since each step depends on the output of the previous one—different time steps and the filtering process can be computed in parallel. The correlation step, however, remains sequential. The structural analysis can also be parallelized, and the so-called "time evolution" simply represents the plotting of results derived from the structural analysis. Once these data are stored within the network structures, they can be quickly retrieved and visualized.

**Section 4.3 limitations:**

What about the limitations concerning the time correlation to identify faults over time, is it always succeeding or are there some cases for which it fails?

The correlation achieves high precision when an appropriate search radius is chosen. The variability in the results reflects the same ambiguities a human interpreter might face when visually comparing two fault networks and deciding whether faults are splitting or merging. The key difference is that, in our approach, these situations are distinguished using a fixed distance criterion, ensuring consistent and reproducible outcomes.

Although the 3 applications proposed are demonstrating several contexts in which the toolbox can be applied, they are all about rifts and extensional systems. What about convergent and strike-slip systems? Do they represent limitations? If yes which ones?

In this paper, we studied the case of extensional systems, but it would be interesting to see the toolbox applied in other settings. In principle, the library can be adapted to a range of tectonic contexts, as the underlying tools are versatile, though the workflow may require some modification depending on the type of data and the tectonic setting. Additional clarifications on this point have been incorporated into the manuscript.

**Minor comments:**

L82: Mattéo et al. (2021). The parenthesis should be in front of the M as the citation is not embedded in the sentence => (Mattéo et al. 2021).

Done.

L96: parenthesis missing before the 2)

Parenthesis added.

L249: there is a parenthesis that should be remove after the "e.g."

Done.

**References:**

All citations below have been reviewed and updated in the manuscript as suggested.

L85: T et al., 2025. It seems that there is a problem with this citation both in text and in the reference section (L615).

L260: References are required for the wet quartzite, wet anorthite and dry olivine flow laws. (Not Neuharth et al., 2022; the actual papers that published the parameters used in the model). L261: "Beneath the lithosphere lies a weak asthenospheric layer composed of wet olivine (Neuharth et al., 2022)"

Neuharth et al., 2022 is not the publication related to the wet olivine flow law. Here, Hirth & Kohlstedt, 2003 should be cited as this is the paper describing the wet olivine flow law used and cited in Neuharth et al., 2022.

Below are some references missing from the reference list (the one I noted, may not be exhaustive):

Panza et al., 2024

Purinton and Bookhagen, 2021

Baker and Wohlenberg, 1971

Canny, 1986

Guo and Hall, 1992

Shmela et al., 2021

Gassmöller et al., 2018

Glerum et al., 2018

Braun and Willett, 2013

Yuan et al., 2019

Saha et al., 2016

Strak and Schellart, 2016

Strak et al., 2011

Willingshofer and Sokoutis, 2009

Philippon et al., 2015

Schlagenhauf et al., 2008

Lathrop et al., 2022

Henza et al., 2010

Jourdon et al., 2025

**Software/Code related remarks:**

The following comments do not need a particular attention from the authors to publish the article but they could take them into account for future development to enhance code visibility, availability, reusability and collaboration.

While cloning the repository, I realized that the repo is 174 MB large, which I agree is not huge, but the code is only made of 6 Python files and a few jupyter notebooks. This size likely comes from the storage of the data used in the jupyter notebooks as tutorials/demonstration. In general, storing large data files in code repository is not recommended as it will impact users when they download, install or update the code from the repository. A better way would be to store those data in another storage place offering long time storage, and link to it to get access to the data.

This suggestion is very useful. The repository will be modified in order to improve the accessibility of the workflows.

In the README it is mentioned that the installation should be done using conda. I understand that it offers a simple way to install your package but it forces the users to use conda. Fortunately, your package can actually be installed without the use of conda, and except for earthpy which can also be installed without conda, there are not that much benefits of strictly restraining the installation to the use of conda. I don't say that you should get rid of it, but I

rather suggest to provide more options to the users. You could consider to add a setup.py to install using pip for instance.

The repository readme will be updated to give miniforge as alternative to Anaconda. Pip alternative will be offered to the user in the future.

---

## Author Comment (AC3)

**RC2: 'Comment on egusphere-2025-3989', Michele Cooke, 12 Oct 2025**

The manuscript presents a new tool, FatBox for delineating faults networks either from the expression of normal faults scarps on topography or from strain maps. The paper describes the modules within FatBox and applies this tool to three different types of normal faults systems to highlight the different types insights that can arise from fault data extracted using Fatbox. As presented, the tool has great potential for normal faults systems. The figures are nicely constructed but the writing can benefit from greater clarity.

- The paper has a strong focus on normal faults within extensional regimes. The paper could either 1) focus only on normal faults and accurately reflect this in the title and throughout the manuscript or 2) broaden the scope to strike-slip and reverse faults by adding examples of these faults and citations to fault detection approaches for those systems.
  - 1. For example, lines 45-47 talk about normal fault growth by linkage. Thes same could be said for strike slip and reverse faults.
  - 2. Strike-slip faults can have geomorphic expression. For example, can FatBox analyze detect strike-slip faulting along pressure ridges? Should present on this around lines 60.
  - 3. Figure 1 only presents normal fault scarp analysis, but FatBox could be applied to dilatational/longitudinal strain maps of reverse faulting or to vorticity/shear strain maps of strike-slip faulting.

This paper investigates in detail the case of extensional systems throughout several applications, but we fully agree that Fatbox can be applied to any type of faults. Other scientists have already begun using the toolbox in different tectonic settings, and their feedback and contributions will help expand the library and demonstrate its versatility. Hence, we prefer to keep the name of the paper identical to the name of the toolbox. Additional details have been added to the Introduction and Methods sections to clarify the range of possible applications.

- 2. The introduction explains how field mapping and numerical models investigation fault network development. The two sentences on physical experiments in the introduction do not adequately explain how these experiments are used to provide insight on fault networks. For example, the paper can explain that within the table-top experiments the material are often scaled so that stress states within the experiment represent the crust. The manuscript has been modified according to the reviewer suggestion to clarify this part.
- 3. Fault connectivity is a tricky element that the process of skeletonization process may or may not do in the same way as a human mapper. Fo example, the faults of Figure 3b maybe be under-connected. How is the correct degree of connectivity assessed?
  The connectivity is based on the distance between features. This connection detection is

performed between the skeletonization and the graph creation stage. Once the network is built as a graph it can be modified easily in the filtering stage (between the extraction and the structural analysis). There, the user can decide to link or unlink structure based on distance, curvature, dip direction, etc. to increase the precision of the network. The final assessment can be made by importing a subset of manually mapped faults, extracting it as a network (tutorial available on Github) and comparing the two datasets. In the end, the user assesses the quality of the fault network extraction. Nevertheless, the idea of "correct degree of connectivity" is a tricky and debatable question, even between experts mapping manually the same area.

**4.** For the reason of c and other issues, the paper would greatly benefit from at least some validation exercise.

Some validations are available in La Rosa et al. 2025. Fatbox was used to extract faults from the DEM of the Afar region and then used to perform an extended structural analysis to compute strain maps. A manually mapped subset was then processed using the automated structural analysis. The strain data from these two datasets were compared and shown to have similar strain values and spatial distribution (see figure S6 from La Rosa et al. 2025). The residuals between the manual and automatic datasets range from  $\pm$  0.1, with just an outlier pixel giving a residual of 0.15. Assuming the manual dataset is representative of 100% of the faults, we calculated that the automatic approach successfully retrieved 93.4% of the total number of faults.

The manuscript has been modified to clarify this point, and detail has been added about the validation.

5.

- Can FatBox map as well as human? Probably not but it certainly creates maps
  faster than a human. Can FatBox save time my providing a map that a human can
  revise more quickly than if they built from scratch?
  Manually mapped datasets can be imported in the workflow and compared to
  semi-automatically extracted networks by Fatbox. Vice versa, networks can be
  exported as shp for example, to be compared with other datasets in GIS software
  or indeed to provide a basis for manual revision. The mention of this possibility
  has been added to the manuscript.
- 2. It is possible that FatBox reduces biases introduced to fault map by human interpretation. The rigorous assessment of that is probably beyond the scope of this study but human mapping bias is well documented in the literature and could be mentioned.

We added some details to clarify the biases.

- 6. Can the analysis work equally well for incremental strain maps and cumulative strain maps?
  - The analysis can be performed for both data sets. It has to be kept in mind though that each approach extracts different entities (all faults for the former case and active faults in the latter case). The availability of these data depends on the kind of model that is analyzed. In this manuscript, incremental strain maps are used in analogue model analysis, while cumulative strain (ie. plastic strain) was used in the numerical model application.
- 7. Some of the fault detection approaches within FatBox (absolute and relative strain threshold and adaptive threshold) have been used before and are not new to this study. Some discussion of the benefits and drawbacks of the different approaches along with citation to previous studies will help the reader make best use of Fatbox.

  We added details to the discussions and references to the other approaches.

**Specific comments:**

- I recommend changing the term 'analog models' to either 'physical experiments' or 'experiments with crustal analog materials'. This change in wording better distinguishes numerical models from laboratory experiments.
   Done.
- The paper uses a lot of passive voice which is not very interesting to read. Try
  adjusting passive voice (e.g. 'fault networks are investigated') to active voice
  ("structural geologists investigate faults networks') and you will find that the text is a
  lot more engaging and direct.
  - We have changed the manuscript accordingly. Thank you for that suggestion.
- When citing a paper that is just one of many examples that you could use, preface the citation with 'e.g.' this signifies that there are many other papers on this topic.
   Throughout the manuscript many citations listings need this e.g. because they do not list all of the papers on that topic.
   Absolutely. We corrected this issue.
- Rewrite the application sub heading include the type of data set analyzed. 3.2
  application 1: topography from Magadi Natron basin; Application 2: strain rate maps
  from numerical models; Application 3: topographic and incremental strain maps from
  experiments
  Done.

Line: 15: "Understanding complex fault networks" is a very vague and your contribution could be more effectively community by being specific.

We changed the vocabulary to use more precise words.

Line 25 (and else where): the word 'graph' is not effectively used here and elsewhere. What do you mean? How is a fault network represented as a 'graph'.

A graph is an ensemble of components mapped at a given time which together compose a network. We updated the manuscript to define the vocabulary early.

Line 31-32: The definition of faults here could equally apply to opening mode joints, veins and dikes, which are certainly not faults. You are missing the key element to faults that is dip or strike slip.

The manuscript has been modified to be more specific.

Line 59: awkward working: Maybe try "...unlike thrust faults within contractional settings where the hanging wall collapses and obscures the fault scarp".

Done.

Lines 95-98: This listing is vague and confusing. Can be clarified by using similar set up for each example 1) uses a topography dataset from Magdi Natron basin, 2) uses a strain rate data set from a numerical model and 3) uses topography and increment strain data sets from experiments. "Fault geometry tracking within an analogue model' doesn't mean anything.

Line 124: This is only for normal fault systems so the text should be revised to state this. Caption modified according to the reviewer suggestion.

Line 128: Throw and heave are displacements. By displacement do you mean net slip? In the paper "displacement" means slip along scarp.

The manuscript has been modified to clarify this point.

Line 150: Incorrect. Particle image velocimetry (PIV) is a process and not a product. What you are extracting the faults from is the strain map. This can be longitudinal strain, shear strain, dilatational strain or vorticity. Thos strains are calculated from the displacement fields generated by PIV. But the fault maps are not extracted from PIV. We changed this misleading sentence.

Line 153: Only in regions with low erosion and deposition rates The manuscript has been modified to clarify this point.

Line 163: Largest continental rift. The mid ocean ridges are larger. We clarified this point.

Line 189: What does 'dedicated functions' mean? Are these bespoke scripts for the particular data set that the user develops?

This sentence refers to functions developed by the authors and already available in the toolbox. Manuscript updated.

Lines 211-212: What does this mean?

This sentence refers to the process of field survey when the geologist walks along the fault and measures the fault structural parameters at regularly spaced distances. We clarified the sentence.

Lines 217: 'Fault displacement and extension'  $\downarrow$  doesn't make sense. I think you mean net slip on the faults? By the way this is one (of many) sentences with passive voice.

This sentence refers to the motion parallel to the fault plane and the associated horizontal displacement (i.e., the amount of extension accommodated by the fault motion). The manuscript has been modified to clarify this point.

Line 219: "accessible though dedicated attributes" This needs more explanation. Absolutely. We added some examples.

Line 320: incorrect PIV is a type of Digital Image Correlation. They are not synonymous. PIV calculates incremental displacement fields. It doesn't calculate velocity because the time step does not emerge from the PIV analysis. The analysis doesn't consider the timing between successive photos.

We corrected the vocabulary.

Line 329: Fault extraction has been performed from experimental strain maps and that literature should be presented because it is not new to this study. The work that I'm most familiar with are for experiments of strike-slip faults. Some studies use a globally set incremental strain threshold (e.g. Hatem et al., 2017 JSG, Visage et al., 2023 Tectonophysics). Some recent papers use a adaptive threshold for fault detection from strain maps (e.g. Chaipornkaew et al., 2022 GRL; Gabriel et al., 2025 Tektonika). In a paper from my research group that is currently under review (under review since July!) we test the sensitivity of the detect strike-slip fault network to adaptive threshold parameters. Seems like this could interest the authors and I am happy to follow up to share the paper.

References to these papers have been added to the manuscript.

Line 371: Extension of the faults? Unclear. Do you mean dip slip along normal faults that accommodate extension? Once again here faults do not have displacement, they have slip as one side of the fault displaces relative to the other.

The manuscript has been modified to clarify the vocabulary.

Line 441: post-extraction validation of the fault network. This paper would be greatly strengthened with a demonstration of validation (see point d). We don't know if the tool is useful if it is not validated. At minimum, this part of the discussion can outline how such a

validation could be performed and the network assessed.

Some detail about the validation performed in La Rosa et al. 2025 has been added to the manuscript. See also details above, in the major comment 4.

Line 470-: The first statement of the conclusions implies that Fatbox provides tools for fault analysis. The tools provided are very helpful for extracting faults, fault tracking and slip information but the analysis itself is done after that data is collected by Fatbox. For example, if someone wanted to analyze the evolution of displacement-length data along a set of faults, Fatbox could extract the data, but the researcher would need to do the analysis of the data. Rewording the text here and in the abstract (lines 27-29) could more accurately convey the value of the tools.

This statement refers to the automated structural analysis tool provided by the toolbox. The kinematic fault parameters, such as extension, displacement (slip along fault) etc, can be computed automatically. Of course, the interpretation and scientific analysis of the resulting plots and data remain the responsibility of the expert user. The manuscript has been modified to clarify this point.